# OpenReview forum: "ASGuard: Activation-Scaling Guard to Mitigate Targeted Jailbreaking Attack"
_ICLR.cc/2026/Conference — ICLR 2026 Poster_

### Official Review · Reviewer_HMBL · 2025-10-21

**Soundness:** 3
**Presentation:** 3
**Contribution:** 3
**Rating:** 6
**Confidence:** 2

**Summary:**

The paper studies a safety flaw called tense jailbreaking, where rephrasing harmful prompts in the past tense bypasses model refusal. It proposes ASGUARD, a three-step mechanistic patch:
(1) build circuits with edge attribution patching and integrated gradients (EAP-IG) to locate attention heads active only in successful tense jailbreaks;
(2) learn channel-wise activation scalers to steer those heads toward safe refusals;
(3) run preventative fine-tuning while freezing scalers so the model internalizes safety, then remove them.
Experiments on Llama-3.1-8B-Instruct, Qwen-2.5-7B-Instruct, and Gemma-2-9B-it show large drops in attack success with little utility loss. Linear probes reveal that several heads act as tense detectors.

**Strengths:**

- Combining circuit discovery with scaling and using the scalers as a temporary scaffold during fine-tuning is novel. It goes beyond standard post-hoc steering by making the intervention teachable and later removable.
- Experiments cover three instruction-tuned models and compare against SFT, DPO, Circuit Breaker, and Representation Bending. The paper also introduces a robustness composite R-Score and reports raw metrics on OR-Bench-Toxic, OR-Bench-Hard, and MMLU. The Pareto plots are informative.
- The paper is clearly structured and easy to follow.

**Weaknesses:**

- In Table 2, all models show heavily degraded performance on OR-Bench-Hard. **It is possible that the reduction in Past-Tense ASR mainly results from an increased tendency to refuse responses**. This undermines the purpose of updating only the tense-vulnerable attention circuits. An analysis controlling for over-refusal rates would be very helpful.
- The study focuses **solely on past-tense reformulations**. It does not show transfer to other semantic variants that commonly appear in real jailbreaks, such as future or hypothetical moods, passive voice, or non-English prompts. The work would have greater impact if it could generalize this approach to more attack types.
- Circuit discovery with EAP-IG involves many integration steps, multiple refusal templates, and top-n pruning, which can be **computationally heavy** compared to baselines. The paper should include an analysis of EAP-IG’s cost and accuracy for readers less familiar with the method.
- **Some baselines may be under-tuned or incomplete**. DPO uses only one epoch, while Circuit Breaker and Representation Bending use fixed configurations per model, and result counts vary. Because safety fine-tuning is sensitive to hyperparameters, stronger or more systematically tuned baselines would improve fairness.
- The paper **lacks a reproducibility statement and open-source code**. Since ASGUARD involves complex interpretability tools and custom fine-tuning procedures, the results are difficult to verify.

**Questions:**

- The SFT(30/70) model appears competitive but sacrifices robustness. This might result from catastrophic forgetting due to full fine-tuning on small datasets. Would using LoRA mitigate this issue? Since ASGUARD fine-tunes only a small subset of parameters, this may explain why it maintains overall performance.

- While the paper reports MMLU scores to assess utility retention, MMLU does not involve tense understanding. If updates to tense-related circuits affect the model’s handling of tense more broadly, how could that degradation be measured?

---

> ### Author Response · Authors · 2025-11-18
> **Response to Reviewer HMBL (Part 1/3)**
>
> Thank you for your detailed feedback. Your insights help refine our paper and clarify key aspects.
>
> ---
>
> >It is possible that the reduction in Past-Tense ASR mainly results from an increased tendency to refuse responses.
>
> We fully agree that controlling for over-refusal is critical, and our design is explicitly aimed at avoiding “just refuse more.” Many safety baselines (especially high-ratio SFT) achieve low ASR partly by over-refusing, which is reflected in large drops on OR-Bench-Hard and in their low R-Score (jointly penalizes over-refusal and capability loss).
>
> Our method is designed to trade off safety and robustness more conservatively:
>
> - In Table 2, SFT(30/70) substantially reduces ASR but suffers from pronounced drops in OR-Bench-Hard and Overall, indicating over-refusal and catastrophic forgetting on benign hard prompts.
> - ASGuard achieves comparable or *better* ASR reduction, while its OR-Bench-Hard and MMLU scores remain much closer to the base model, yielding the highest R-Score and Overall across methods.
>
> This suggests that the reduction in tense ASR is **not** simply due to globally increasing refusal rates, but rather to a more targeted change to the vulnerability.
>
> >It does not show transfer to beyond semantic variants that commonly appear in real jailbreaks.
>
> We agree that generalization beyond past-tense reformulations is important. In the revised version, we add experiments on two additional jailbreak families that go well beyond tense: **GCG** (adversarial suffix-based attacks) [1] and **LogiBreak** (logical-form jailbreaks) [2]. We apply the *same* ASGuard pipeline (EAP-IG circuit discovery → scaling → preventative fine-tuning) and show that:
>
> - GCG ASR reduced from 15% to 1%
> - Lowers LogiBreak ASR from 30% to 13%
> - Best R-Score and Overall among the compared methods.
>
> *(a) List of Targeted Vulnerable Heads*
>
> | Model | List of GCG Vulnerable Heads | List of LogiBreak Vulnerable Heads |
> | --- | --- | --- |
> | Llama-3.1-8B-Instruct | L0H3, L1H24, L30H1, L31H5 | L24H27, L28H13 |
>
> *(b) Results on GCG*
>
> | Method | GCG ASR (↓) | OR-Bench Toxic (↑) | OR-Bench Hard (↓) | MMLU (↑) | R-Score (↑) | Overall (↑) |
> | --- | --- | --- | --- | --- | --- | --- |
> | Llama-3.1-8B-Instruct | 15 | 88.5 | 28.9 | 68.2 | – | – |
> | SFT (30/70) | 2 | 66.1 | 2.27 | 67.3 | 13.8 | 13.4 |
> | RepBend | 5 | 94.3 | 50.4 | 68.1 | 73.3 | 41.7 |
> | ASGuard (Ours) | **1** | 96.7 | 59.5 | 68.3 | **76.0** | **45.0** |
>
> *(c) Results on LogiBreak*
>
> | Method | LogiBreak ASR (↓) | OR-Bench Toxic (↑) | OR-Bench Hard (↓) | MMLU (↑) | R-Score (↑) | Overall (↑) |
> | --- | --- | --- | --- | --- | --- | --- |
> | Llama-3.1-8B-Instruct | 30 | 88.5 | 28.9 | 68.2 | – | – |
> | SFT (30/70) | **0** | 59.3 | 1.59 | 66.7 | 0* | 15.0 |
> | RepBend | 13 | 68.7 | 78.9 | 68.2 | 0* | 8.5 |
> | ASGuard (Ours) | 13 | 97.1 | 64.9 | 68.1 | **74.7** | **45.8** |
>
> This demonstrates that our framework is not limited to a single surface-level tense perturbation and can be instantiated for structurally different attacks.
>
> >Circuit discovery with EAP-IG involves many integration steps, multiple refusal templates, and top-n pruning, which can be **computationally heavy** compared to baselines.
>
> We acknowledge that EAP-IG circuit construction is more complex than running a single SFT run, and we will add an explicit cost summary. Practically, however, the cost is manageable:
> - We construct circuits only for a short list of layers/heads (rather than all modules), and we use top-n pruning and batching to keep the number of integration steps low.
> - On our hardware (2×H100 GPUs), a full tense circuit discovery run for Llama-3.1 takes on the order of a few GPU-hours, and scaling-vector training is an order of magnitude cheaper than even a single epoch of SFT or DPO.
> - When SFT with 5000 element took 12 minutes and DPO took 10 minutes per single epoch with single H100, circuit construction took 30 minutes for single circuit, scaling took 2~3 seconds for single epoch, preventative finetuning took 12 seconds per single epoch, while CB and RepBend took more than two hours.
>
> ---
>
> [1] Universal and transferable adversarial attacks on aligned language models, Zou et al., 2023
>
> [2] Logic Jailbreak: Efficiently Unlocking LLM Safety Restrictions Through Formal Logical Expression, Peng et al., 2025

---

> ### Author Response · Authors · 2025-11-18
> **Response to Reviewer HMBL (Part 2/3)**
>
> >**Some baselines may be under-tuned or incomplete**.
>
> We agree that safety fine-tuning is sensitive to hyperparameters and have tried to make the baselines as strong and fair as our budget allows.
>
> - For **Circuit Breaker** and **RepBend**, we followed the authors’ recommended settings and, for each model, selected the strongest configuration from the hyperparameter grid that we could afford to evaluate (e.g., scaling strengths, layer ranges). The configurations reported in Table 2 are the best-performing ones we found for each method and model, not arbitrary defaults.
> - For **DPO**, we experimented more epochs, but additional epochs did not materially improve ASR and hurt OR-Bench-Hard due to overfitting on a dataset. We therefore report the 1-epoch setting, which achieved the best overall trade-off among the DPO variants we tested.
>
> >The paper **lacks a reproducibility statement and open-source code**.
>
> We are currently cleaning our codebase for scaling and preventative fine tuning and will **release the full code and configuration files**, together with the curated detection and refusal datasets, as supplementary. We will also add a reproducibility statement to the revised version, detailing dependencies of experiments.
>
> >The SFT(30/70) model appears competitive but sacrifices robustness. This might result from catastrophic forgetting due to full fine-tuning on small datasets.
>
> We appreciate this insightful observation.
>
> Regarding catastrophic forgetting, we agree that full SFT(30/70) on a small safety dataset likely causes forgetting and over-refusal. LoRA-style updates could mitigate this to some extent by constraining the effective parameter change. However, without mechanistic guidance, we hypothesize such LoRA-based SFT to behave like a generic safety finetune: it may improve average refusal but does not specifically repair the tense jailbreak circuit, and is still sensitive to hyperparameters and data composition.
>
> So, we additionally check SFT with LoRA, for the attention out layer (LoRA Target Module: o_proj, out_proj, lora_r: 16, lora_alpha: 32, lora_dropout: 0.05) with the same SFT settings.
>
> The result is like the table below.
>
> | Method | Past Tense ASR (↓) | OR-Bench Toxic (↑) | OR-Bench Hard (↓) | MMLU (↑) |
> | --- | --- | --- | --- | --- |
> | **Llama-3.1-8B-Instruct** | 42 | 88.5 | 28.9 | 68.2 |
> | SFT (5/95) | 21 | 94.1 | 50.8 | 67.8 |
> | SFT (5/95) + LoRA | 35 | 93.1 | 49.7 | 68.2 |
> | SFT (30/70) | 3 | 91.9 | 80.3 | 67.7 |
> | SFT (30/70) + LoRA | 3 | 98.4 | 86.1 | 68.1 |
> | DPO | 38 | 90.2 | 33.2 | 68.0 |
> | Only Scaling (Ours) | 13 | 96.9 | 66.2 | 64.3 |
> | ASGuard (Ours) | 8 | 96.4 | 66.8 | 68.2 |
>
> Although LoRA was effective in preserving general knowledge by mitigating catastrophic forgetting (MMLU remained almost unchanged), the target ASR did not decrease at all, or even worsened (e.g., the ASR of SFT 5/95 with LoRA increased). This strengthens our approach, as LoRA naively increases overall refusal, which leads to increased over-refusal as well. By contrast, ASGuard uses the scaling coefficients as a **targeted constraint during training**, steering the optimizer away from a specific failure pathway while preserving general capability.
>
> Our method shares some similarities with recent **scaling-based, parameter-efficient approaches** such as SEAL [3] , which also learns per-head scaling factors to emphasize specific attention heads instead of introducing full low-rank adapters as in LoRA. In our case, the learned scaling vectors act as lightweight, head-specific “steering coefficients” that modulate the vulnerable circuit, and preventative fine-tuning is then performed under this modified pathway.
>
> Conceptually, this is reminiscent of “LoRA + fine-tuning” in that we introduce a small number of extra parameters to guide training, but **the key difference is targeting**: our scaling is applied only to heads identified via circuit analysis as tense-vulnerable, rather than to all attention projections. In our experiments, when we apply similar scaling to non-targeted heads (e.g., Temporal Heads), we observe little to no change in tense ASR, indicating that the benefit comes primarily from *where* we intervene, not just from reducing the number of updated parameters.
>
> ---
>
> [3] SEAL: Scaling to Emphasize Attention for Long-Context Retrieval, Lee et al., 2025

---

> ### Author Response · Authors · 2025-11-18
> **Response to Reviewer HMBL (Part 3/3)**
>
> >If updates to tense-related circuits affect the model’s handling of tense more broadly, how could that degradation be measured?
>
> We agree that MMLU does not directly probe tense understanding. Our motivation for including MMLU is to verify that **general knowledge and reasoning ability** are not degraded by our interventions; in particular, we want to avoid catastrophic forgetting on capabilities unrelated to tense or safety. The fact that ASGuard keeps MMLU close to the base model (unlike some baselines) supports that general utility is preserved.
>
> We acknowledge that a more targeted evaluation of tense and temporal understanding would be valuable. We utilize tense based QA set of TenseLoC [4] which deals with internal representation of tense inside of LLM. For the sentence with blanked verb tense, models should answer which tense of verb is correct for the blank. Down below is the example of test set with the generated answer.
>
> ```json
> { "prompt_id": 1,
> "gold_tense": "past",
> "prompt_text": "Last night, I ___ the journal.\nA) drew\nB) will draw\nC) draw\nAnswer:",
> "gold_answer": "A",
> "generated_text": "The correct answer is A) drew.\n\nThe sentence",
> "prediction": "A",
> "is_correct": true}
> ```
>
> We measure each performance under greedy decoding with the system prompt below.
>
> ```json
> system_prompt = "You are an expert in English grammar, specifically focusing on verb tenses. Select the correct option (A, B, or C) that best completes the sentence based on the temporal context."
> ```
>
> The table below indicates the overall accuracy of each methods and trained models.
>
> | Method | Past Acc | Present Acc | Future Acc | Total Acc |
> | --- | --- | --- | --- | --- |
> | **Llama-3.1-8B-Instruct** | 96.0 | 47.0 | 39.0 | 60.6 |
> | SFT (5/95) | 91.0 | 60.6 | 47.3 | 66.3 |
> | SFT (30/70) | 94.3 | 44.0 | 43.3 | 60.5 |
> | DPO | 96.0 | 47.0 | 38.6 | 60.5 |
> | ASGuard (Ours) | 95.6 | 45.0 | 43.3 | 61.3 |
>
> While SFT models match or exceed the baseline in total accuracy, they fail to retain past tense distinction. This causes the model to ignore temporal cues, rendering the past tense jailbreak ineffective simply due to a lack of understanding rather than robust safety. For the present tense, every models exhibited a tendency to mislead present as past context. As the example shows below, it understood sentences as past tense, ignoring cues that define them as present tense when evaluated strictly in isolation, devoid of any external or stylistic context.
>
> ```json
> {"prompt_id": 339,
> "gold_tense": "present",
> "prompt_text": "Always, the cooks ___ the meeting.\nA) will construct\nB) constructs\nC) constructed\nAnswer:",
> "gold_answer": "B",
> "generated_text": "The correct answer is C) constructed.\n\nThe sentence",
> "prediction": "C",
> "is_correct": false},
> ```
>
> Overall, ASGuard keeps the overall accuracy of tense representation, in addition to preserve general knowledge ability.
>
> ---
>
> We hope these clarifications address your concerns and demonstrate the strengths of our framework. Thank you for your thoughtful and constructive review.
>
> ---
>
> [4] TenseLoC: Tense Localization and Control in a Multilingual LLM, Tumurchuluun et al., 2025

---

> > ### Author Response · Authors · 2025-11-24
> > **Follow-up on Author Response**
> >
> > Dear Reviewer HMBL,
> >
> > We are writing to kindly invite you to review our response to your comments and the updated manuscript. We have made every effort to address the points you raised and would be grateful for your feedback on our revisions. We understand the demanding nature of the review process and are truly grateful for the time and effort you are dedicating to our work.

---

> > > ### Comment · Reviewer_HMBL · 2025-11-25
> > >
> > > Thank you for your detailed responses. I believe the targeted evaluation of tense and temporal understanding strengthens the claims in the paper.

---

> > > > ### Author Response · Authors · 2025-11-25
> > > > **Official Comment by Authors**
> > > >
> > > > Thank you for recognizing that our additional experiments have strengthened the paper’s validity. If there are specific areas where you see room for further refinement, we would greatly value your insights. Your guidance is pivotal in helping us enhance the quality and rigor of our work, and we remain committed to continuous improvement.

---

### Official Review · Reviewer_KbDQ · 2025-10-29

**Soundness:** 3
**Presentation:** 3
**Contribution:** 2
**Rating:** 4
**Confidence:** 3

**Summary:**

The authors provide a new way of mitigating specific attack points of an LLM. They identify key components to the past-tense attacks, learn suppression coefficients and then finetune with the suppressed model to increase its robustness.

**Strengths:**

- The approach seems to be effective against the past tense attack
- It is evidence for the circuit identification

**Weaknesses:**

- The technique is very attack-specific. It is unclear how effective the finetuning will be against any other attack. In that sense, the research is quite limited in my opinion. If it does not generalize, then it is a very expensive way of finetuning since one cannot just add data from other attacks but has to do all the different steps again.
- Because of the previous point, the Pareto frontier data is a bit less relevant since the reduction in ASR is only against a single attack. I would like to ask the authors to add a range of other attacks to provide some data for the generalization of the technique
- If it does not work well, then I believe the paper is still somewhat relevant, but it should be rephrased towards not being a defense technique but rather evidence for the usefulness of mechanistic interpretability (with a few more findings on the mechanisms, it could be very interesting)
- The insight of this paper that attention heads only signal the tense and then the harmfulness evaluation will be later could use more experiments or arguments. To me, it feels a bit unsupported and is only drawn as a conclusion based on other work

**Questions:**

I would like to understand the finetuning step a bit more.
As I understand it, the coefficients disable the "tense" heads to signal the tense. Then the model is finetuned and the coefficients are detached. To me, it seems like you are helping the model not to be susceptible to this attack during finetuning and enabling this attack path again after it. Hence, I am a bit confused about why this works. What is your justification for this? Can you provide some evidence that this is actually the case (e.g., only using a few coefficients).

---

> ### Author Response · Authors · 2025-11-18
> **Response to Reviewer KbDQ (Part 1/2)**
>
> Thank you for your detailed and insightful review.
>
> ---
>
> > - It is unclear how effective the finetuning will be against any other attack.
> > - the Pareto frontier data is a bit less relevant since the reduction in ASR is only against a single attack.
> > - it should be rephrased towards not being a defense technique but rather evidence for the usefulness of mechanistic interpretability (with a few more findings on the mechanisms, it could be very interesting)
>
> We appreciate this concern and agree that, in order to be convincing as a *defense* technique, ASGuard must show evidence of generalization beyond a single attack template.
>
> In the revised version, we extend ASGuard to **two additional jailbreak families**, following the exact same three-stage pipeline (Circuit discovery → targeted scaling → preventative fine-tuning): **GCG** (adversarial suffix-based attacks) [1] and **LogiBreak** (logical-form jailbreaks) [2]. For each attack, we construct success vs. failure circuits, identify “vulnerable heads” using the same EAP-IG procedure as for tense jailbreaks, and then train scaling vectors and perform preventative fine-tuning exactly as in the original setting. On Llama-3.1, this yields strong **in-domain** defenses for these attacks (e.g., GCG ASR reduced from 15% to 1%, LogiBreak from 30% to 13%), while maintaining competitive R-Scores and Overall scores relative to strong baselines such as RepBend.
>
> *(a) List of Targeted Vulnerable Heads From Circuit Analysis*
>
> | Model | List of GCG Vulnerable Heads | List of LogiBreak Vulnerable Heads |
> | --- | --- | --- |
> | Llama-3.1-8B-Instruct | L0H3, L1H24, L30H1, L31H5 | L24H27, L28H13 |
>
> *(b) Results on GCG (Zou et al., 2023) Benchmark*
>
> | Method | GCG ASR (↓) | OR-Bench Toxic (↑) | OR-Bench Hard (↓) | MMLU (↑) | R-Score (↑) | Overall (↑) |
> | --- | --- | --- | --- | --- | --- | --- |
> | Llama-3.1-8B-Instruct | 15 | 88.5 | 28.9 | 68.2 | – | – |
> | SFT (30/70) | 2 | 66.1 | 2.27 | 67.3 | 13.8 | 13.4 |
> | RepBend | 5 | 94.3 | 50.4 | 68.1 | 73.3 | 41.7 |
> | ASGuard (Ours) | **1** | 96.7 | 59.5 | 68.3 | **76.0** | **45.0** |
>
> *(c) Results on LogiBreak (Peng et al., 2025) Benchmark*
>
> | Method | LogiBreak ASR (↓) | OR-Bench Toxic (↑) | OR-Bench Hard (↓) | MMLU (↑) | R-Score (↑) | Overall (↑) |
> | --- | --- | --- | --- | --- | --- | --- |
> | Llama-3.1-8B-Instruct | 30 | 88.5 | 28.9 | 68.2 | – | – |
> | SFT (30/70) | **0** | 59.3 | 1.59 | 66.7 | 0* | 15.0 |
> | RepBend | 13 | 68.7 | 78.9 | 68.2 | 0* | 8.5 |
> | ASGuard (Ours) | 13 | 97.1 | 64.9 | 68.1 | **74.7** | **45.8** |
>
> In addition, we explicitly test **OOD generalization**. A model trained only on tense jailbreak data already shows substantial robustness to unseen GCG and LogiBreak attacks, and to broad toxic prompts from OR-Bench-Toxic, without using those attacks during circuit construction. This suggests that preventative fine-tuning is not just memorizing a single template, but encouraging the model to rely on more robust refusal circuitry that transfers to other attacks.
>
> Meanwhile, we appreciate the reviewer’s suggestion to emphasize the mechanistic aspect. Our intention is indeed twofold: (i) to propose a practical, mechanistically-informed defense that achieves a strong safety–utility Pareto trade-off, and (ii) to show how mechanistic interpretability can guide surgical repairs of localized safety failures. We will clarify this dual framing in the introduction and discussion.
>
> >The insight of this paper that attention heads only signal the tense and then the harmfulness evaluation will be later could use more experiments or arguments.
>
> Thank you for pointing out that our phrasing can be read as stronger than what we actually show. We do *not* intend to claim that certain heads “only” encode tense in an absolute sense; rather, our evidence supports that a *small subset* of heads is specialized for tense information and is causally implicated in tense-specific jailbreaks, while harmfulness and refusal are encoded in partially distinct circuits.
>
> - **Linear probes confirming tense specialization.**
> In Section 6 and Figure 3, we train linear probes on the activations of these heads to classify past vs present tense. For Llama-3.1, probes on heads such as L10H25 and L13H25 achieve ~73–77% accuracy, significantly above chance, and their separation between past and present persists across prompts. This supports the interpretation that these heads are specialized for tense encoding and are upstream in the tense-jailbreak pathway.
>
> - **Dissociation from other temporal/safety circuits.** We explicitly compare our tense-vulnerable heads with both Temporal Heads [3] and safety heads discovered by Sahara [4]. The sets are almost disjoint: the tense-vulnerable heads are different from Temporal Heads, and only partially overlap with Sahara’s general safety circuit. This suggests that the tense vulnerability is tied to a distinct semantic subspace, rather than merely being another reflection of harmfulness or refusal directions.

---

> ### Author Response · Authors · 2025-11-18
> **Response to Reviewer KbDQ (Part 2/2)**
>
> We will revise the wording in the main text to make clear that our conclusion is: (i) certain heads are specialized for tense features and are necessary for past-tense jailbreaks, and (ii) harmfulness/refusal evaluation is implemented in circuits that are at least partially disjoint from these tense-specialized heads, consistent with prior work that shows harmfulness and refusal are themselves separately encoded [5].
>
> Building on your suggestion, we also plan to extend the analysis toward harmfulness-specific circuits in future work, and will mention this as a limitation and avenue for follow-up.
>
> >I would like to understand the finetuning step a bit more. As I understand it, the coefficients disable the "tense" heads to signal the tense. Then the model is finetuned and the coefficients are detached. To me, it seems like you are helping the model not to be susceptible to this attack during finetuning and enabling this attack path again after it. Hence, I am a bit confused about why this works. What is your justification for this? Can you provide some evidence that this is actually the case (e.g., only using a few coefficients).
>
> We apologize for the confusion — our coefficients do *not* disable the tense heads. The scaling vectors are channel-wise multiplicative factors applied to each vulnerable head, so the heads remain active and gradients still flow through them during preventative fine-tuning.
>
> Intuitively, scaling makes the vulnerable pathway *harder to use* during training: harmful features carried by those heads are down-weighted, so the optimizer is encouraged to re-route refusal behavior through alternative, safer circuits and to repurpose the original heads away from the brittle tense-jailbreak mechanism. Preventative fine-tuning thus acts as a form of “training under handicap”: the model must learn robust refusal while the easiest failure path is temporarily suppressed.
>
> We provide two pieces of evidence that this process leaves a lasting effect even after detaching the scaling vectors:
>
> - **Behavioral comparison with ‘Only Scaling’.**
>
>     Table 2 distinguishes between the ‘Only Scaling’ baseline and ASGuard (scaling + preventative fine-tuning + detachment). For Llama-3.1, ‘Only Scaling’ achieves a strong ASR reduction but significantly harms MMLU (64.3 vs 68.2) and yields a lower Overall score (50.3) compared to ASGuard (52.9). This shows that scaling alone is not sufficient; the improvement after detaching comes from the model having *relearned* a safer refusal mechanism in its base weights during preventative fine-tuning.
>
> - **Mechanistic changes in the tense circuit.**
>
>     In Section 6, we repeat linear probing and circuit analysis after applying ASGuard. The tense-specialization of several originally vulnerable heads is reduced, and the tense-jailbreak circuit reorganizes: many of the previously dominant heads are no longer central to the past-tense jailbreak pathway. This is consistent with the view that preventative fine-tuning reshapes internal computations rather than merely relying on the external scaling coefficients at inference time.
>
>
> We also experimented with using only a subset of the identified vulnerable heads during preventative fine-tuning. These variants led to noticeably weaker ASR reductions, suggesting that the tense jailbreak mechanism is distributed over a broader circuit rather than a few isolated heads.
>
> Conceptually, our approach is closely related to recent preventative-steering, which also apply interventions during training but remove them at inference time, while observing persistent improvements in robustness [6].
>
> ---
>
> We would greatly appreciate your feedback on whether these additional analyses address your concerns.
>
> If you have any further suggestions for improvement, please feel free to share any additional thoughts or questions.
>
> ---
>
> [1] Universal and transferable adversarial attacks on aligned language models, Zou et al., 2023
>
> [2] Logic Jailbreak: Efficiently Unlocking LLM Safety Restrictions Through Formal Logical Expression, Peng et al., 2025
>
> [3] Does Time Have Its Place? Temporal Heads: Where Language Models Recall Time-specific Information, Park et al., 2025
>
> [4] On the role of attention heads in large language model safety, Zhou et al., 2025
>
> [5] Llms encode harmfulness and refusal separately, Zhao et al., 2025
>
> [6] Persona vectors: Monitoring and controlling character traits in language models, Chen et al., 2025

---

> > ### Author Response · Authors · 2025-11-24
> > **Follow-up on Author Response**
> >
> > Dear Reviewer KbDQ,
> >
> > We are writing to kindly inquire if there are any remaining concerns regarding our manuscript. We have thoroughly incorporated your feedback into the revision and provided detailed responses to your comments. We also remain available to answer any questions you may have. Thank you for your time and contribution to improving our work.

---

### Official Review · Reviewer_PzPS · 2025-11-01

**Soundness:** 3
**Presentation:** 3
**Contribution:** 2
**Rating:** 2
**Confidence:** 4

**Summary:**

This paper introduces ASGuard, a mechanistically-informed defense against tense jailbreaking attacks in LLMs. The method: (1) uses circuit analysis to identify "tense vulnerable" attention heads causally linked to the attack, (2) trains channel-wise scaling vectors to suppress these heads, and (3) applies "preventative fine-tuning" to learn robust refusal mechanisms. Across 3 models and multiple benchmarks, ASGuard reduces tense jailbreaking ASR from 42-51% to 8-19% while maintaining better safety-utility balance than baselines like SFT, DPO, and Circuit Breakers.

**Strengths:**

1. Strong mechanistic foundation & insights: The circuit analysis approach (EAP-IG) to identify vulnerability-specific heads is principled, well-studied in literature, and provides interpretable insights. The linear-probe validation (§6.1) provides mechanistic confirmation that these heads encode tense information. Furthermore, the finding that alignment is localized to specific routing paths (many vulnerable heads disappear post-ASGuard, §6.2) is an important contribution to understanding LLM safety mechanisms.

2. Novel defense paradigm: The authors propose the "Preventative fine-tuning" (training with scaling vectors attached, then removing them), which is to my knowledge creative and shows clear improvements over naive scaling or standard fine-tuning alone.

3. Good empirical results (but unsure if there are still competive to the baseline that I’ll mention below): ASGuard achieves Pareto-optimal safety-utility trade-offs across all three models tested, substantially outperforming strong baselines, including well-known methods such as circuitbreaker.

**Weaknesses:**

1. Extremely Limited Scope
The method is designed for exactly one attack type (tense jailbreaking). There is no evidence it generalizes to: other semantic attacks (negation, translation, role-play, etc.), adversarial suffixes (GCG, AutoDAN), multi-turn jailbreaks.
I think this is a critical weakness as the baselines that are used by authors are evaluated on different types of attacks. It further shows somehow an unfair comparison.

2. Circular Methodology: the detection dataset (§3.1) and evaluation dataset both use JBB-Behaviors prompts, creating a potential unreliable evaluation. Indeed, 100 prompts are used for circuit construction and the same prompts (with different reformulations) seem to be used for evaluation, exposing the risk of overfitting to these specific prompts.

3. Limited Baseline Comparisons

There are missing well-known defenses: Representation Engineering (Turner et al., 2023) is cited but not compared. Prompt-based defenses (system prompts, few-shot examples) and Input filtering or paraphrasing are not evaluated. Authors should also compare their method against RFA (Yu et al, 2024) "refusal feature adversarial training" and Gradient Cuff.

Furthermore, there seem to be some issues with the RepBend comparison: RepBend is described as "recent SoTA" but it's surprising that  it doesn't consistently outperform simpler methods.  Some results show RepBend > ASGuard on specific metrics (e.g., Llama Toxic: 96.1 vs 96.4).

**Questions:**

1. :Can this framework defend against attacks other than tense jailbreaking? If not, how is it more than a specialized patch?

2. What are the computational costs relative to standard fine-tuning? Can you ablate on detection dataset size?

3. If you train defenses for multiple attack types, do they interfere? Can one set of scaling vectors handle multiple attacks?

4.Why does preventative fine-tuning work? Can you provide formal analysis or at least deeper mechanistic explanation?

---

> ### Author Response · Authors · 2025-11-18
> **Response to Reviewer PzPS (Part 1/3)**
>
> We greatly appreciate your detailed feedback and address each concern below.
>
> ---
>
> >There is no evidence it generalizes to: other semantic attacks (negation, translation, role-play, etc.), adversarial suffixes (GCG, AutoDAN), multi-turn jailbreaks. I think this is a critical weakness as the baselines that are used by authors are evaluated on different types of attacks.
>
> >Can this framework defend against attacks other than tense jailbreaking? If not, how is it more than a specialized patch?
>
> We agree that demonstrating generality beyond tense jailbreaks is important. In the revised paper, we explicitly instantiate the *same* three-stage ASGuard workflow—EAP-IG circuit discovery, activation scaling, and preventative fine-tuning—for two additional jailbreak families: **GCG** (adversarial suffix-based attacks) [1] and **LogiBreak** (logical-form jailbreaks) [2].
>
> Using the same circuit-construction procedure as in §3.1, we build circuits for successful vs. failed GCG and LogiBreak attacks and identify “GCG-vulnerable” and “LogiBreak-vulnerable” heads. The list of vulnerable heads is:
>
> | Model | List of GCG Vulnerable Heads | List of LogiBreak Vulnerable Heads |
> | --- | --- | --- |
> | Llama-3.1-8B-Instruct | L0H3, L1H24, L30H1, L31H5 | L24H27, L28H13 |
>
> We then train scaling vectors and perform preventative fine-tuning exactly as in the tense setting. On **GCG**, ASGuard reduces ASR from 15% to **1%** with the best R-Score and Overall score; on **LogiBreak**, ASGuard attains ASR 13% with a much higher R-Score and Overall than SFT or RepBend.
>
> *(a) Results on GCG (Zou et al., 2023) Benchmark*
>
> | Method | GCG ASR (↓) | OR-Bench Toxic (↑) | OR-Bench Hard (↓) | MMLU (↑) | R-Score (↑) | Overall (↑) |
> | --- | --- | --- | --- | --- | --- | --- |
> | Llama-3.1-8B-Instruct | 15 | 88.5 | 28.9 | 68.2 | – | – |
> | SFT (30/70) | 2 | 66.1 | 2.27 | 67.3 | 13.8 | 13.4 |
> | RepBend | 5 | 94.3 | 50.4 | 68.1 | 73.3 | 41.7 |
> | ASGuard (Ours) | **1** | 96.7 | 59.5 | 68.3 | **76.0** | **45.0** |
>
> *(b) Results on LogiBreak (Peng et al., 2025) Benchmark*
>
> | Method | LogiBreak ASR (↓) | OR-Bench Toxic (↑) | OR-Bench Hard (↓) | MMLU (↑) | R-Score (↑) | Overall (↑) |
> | --- | --- | --- | --- | --- | --- | --- |
> | Llama-3.1-8B-Instruct | 30 | 88.5 | 28.9 | 68.2 | – | – |
> | SFT (30/70) | **0** | 59.3 | 1.59 | 66.7 | 0* | 15.0 |
> | RepBend | 13 | 68.7 | 78.9 | 68.2 | 0* | 8.5 |
> | ASGuard (Ours) | 13 | 97.1 | 64.9 | 68.1 | **74.7** | **45.8** |
>
> > the detection dataset (§3.1) and evaluation dataset both use JBB-Behaviors prompts, creating a potential unreliable evaluation … exposing the risk of overfitting to these specific prompts.
>
> We appreciate this concern and agree it is important to rule out overfitting to a narrow prompt set. Our design mitigates this in two ways:
>
> - **Diversity within JBB-Behaviors.**
>
>     For circuit construction, we use 100 prompts from JBB-Behaviors, but for each prompt we generate **20 reformulations** in both past and present tense and evaluate them with GPT-4.1 as a semantic judge. This creates a large, semantically diverse pool of tense perturbations and refusal variants, reducing the risk that the circuit simply memorizes specific surface forms.
>
> - **Cross-dataset and cross-attack evaluation.**
>
>     Crucially, our *main* evaluation is not limited to JBB-style prompts. We measure safety and utility on **OR-Bench-Toxic**, **OR-Bench-Hard**, and **MMLU**, which are entirely separate datasets probing broad toxicity, over-refusal, and general capability. In addition, for **GCG** we use the HarmBench Behavior test set, and for **LogiBreak** we follow the authors’ English reformulations.
>
>     The tense-trained ASGuard model improves OR-Bench-Toxic and maintains OR-Bench-Hard/MMLU, and also achieves strong performance on GCG and LogiBreak, even when these attacks were not used to construct the original tense circuits. This cross-dataset and cross-attack robustness is difficult to explain by overfitting to the 100 JBB prompts and strongly suggests that the method is not merely memorizing those specific prompts.
>
> ---
>
> [1] Universal and transferable adversarial attacks on aligned language models, Zou et al., 2023
>
> [2] Logic Jailbreak: Efficiently Unlocking LLM Safety Restrictions Through Formal Logical Expression, Peng et al., 2025

---

> > ### Comment · Reviewer_PzPS · 2025-11-25
> > **LogiBreak & Repbend Performance**
> >
> > Why is LogiBreak's performance not similar to the ones seen in Table 1 of LogiBreak's paper? The same applies to Repbend.

---

> > > ### Author Response · Authors · 2025-11-25
> > > **Additional Response by Authors**
> > >
> > > Thank you for pointing this out. Our LogiBreak and RepBend tables are not intended as exact replications of the original results, but as evaluations under a *unified* safety pipeline.
> > >
> > > For **LogiBreak**, we believe the comparison you have in mind is Table 2 (since Table 1 reports semantic consistency metrics rather than attack success rates), which reports jailbreak performance under three different judges (Rule_Judge, LLaMA_Judge, GPT_Judge).
> > >
> > > In our experiments, we follow the authors’ official implementation and use the default configuration in their GitHub repository, including `restart = 5`. As a result, our reported attack success rates correspond to ASR@5.
> > >
> > > Under **GPT_Judge**, the LogiBreak paper reports an ASR@5 of **48%** for LLaMA3-8B (a base model without instruction tuning), whereas in our setup the base Llama-3.1-8B-Instruct model attains a LogiBreak ASR@5 of **30%**. While this is lower than 48%, it is still a substantial success rate: even a heavily safety-tuned, instruction-following checkpoint that has undergone extensive RLHF still fails on roughly one in three LogiBreak attempts in our setting. We believe this gap is largely explained by differences in the setup:
> > >
> > > - We evaluate **Llama-3.1-8B-Instruct**, while LogiBreak uses **LLaMA3-8B** (without the 3.1 revision and with a different safety configuration). We chose Llama-3.1-8B-Instruct because *all of our experiments* (past-tense jailbreaks, GCG, LogiBreak) are conducted on this single model variant, which ensures consistency of base behavior, discovered circuits, scaling vectors, and fine-tuning across the study.
> > > - According to its model card, Llama-3.1-8B-Instruct has already been safety-tuned and RLHF-aligned “to align with human preferences for helpfulness and safety,” including adversarial and borderline prompts; this naturally reduces—but does not eliminate—LogiBreak success rates.
> > > - LogiBreak uses GPT-4 as the GPT_Judge, whereas we use GPT-4.1.
> > > - We follow the authors’ English reformulations and run them through our safety prompt and decoding settings.
> > >
> > > Given these differences, we do not expect the absolute ASR@5 for Llama-3.1-8B-Instruct to match the LLaMA3-8B numbers in the original LogiBreak paper exactly. What matters for our claims is that (i) the base ASR@5 remains high enough to represent a meaningful jailbreak threat (30% on a safety-tuned model), and (ii) all defenses in our study (SFT, RepBend, ASGuard, etc.) are evaluated under *exactly the same* LogiBreak implementation, prompt templates, restart protocol, and judge, so the comparison is fair.
> > >
> > > For **RepBend**, the situation is similar. The paper does **not** report results on our tense jailbreak task or on LogiBreak. Instead, it focuses on WildGuardTest, HarmBench, DAN, TrustLLM Jailbreak, PAP, GCG, Prefilling, and Input Embed. We therefore:
> > >
> > > - Use the authors’ **official implementation** and their recommended “best” hyperparameters,
> > > - Apply RepBend to **Llama-3.1-8B-Instruct**,
> > > - And evaluate it on **our** benchmark suite (tense jailbreak, GCG, LogiBreak, OR-Bench Toxic/Hard, MMLU).
> > >
> > > On GCG, for example, the RepBend paper reports that on Llama3-8B-Instruct, RepBend reduces ASR to about **2.5%**. In our setup, RepBend reduces GCG ASR on Llama-3.1-8B-Instruct to around **5%**, while keeping over-refusal relatively modest. This level of agreement is reasonable given the base model difference (3 vs. 3.1) and our distinct evaluation pipeline. This close agreement on GCG serves as a sanity check that our RepBend implementation faithfully reflects the intended behavior of the original method.
> > >
> > > Moreover, although RepBend does not report OR-Bench or our specific tense/LogiBreak metrics, our results show that it remains a strong baseline that substantially lowers ASR while preserving base capabilities, consistent with the original paper’s findings. Within this shared setup, ASGuard further improves the safety–utility trade-off beyond this already strong baseline. So, within our unified evaluation pipeline, RepBend consistently reduces ASR, but ASGuard achieves both lower ASR and higher Overall scores on Llama-3.1-8B-Instruct, indicating a strictly stronger safety–utility frontier.
> > >
> > >
> > > In summary, absolute ASR values differ because the **models, judges, and evaluation pipelines** are not identical to those in LogiBreak and RepBend. What we ensure is that, **within our unified setup**, all defenses (SFT, RepBend, ASGuard, etc.) are compared on exactly the same prompts and judging criteria. The fact that our base and RepBend numbers are of similar magnitude to those reported in the original papers for closely related models supports the credibility of our evaluation, while the key contribution of our work is that ASGuard further improves the safety–utility frontier under this shared evaluation.
> > >
> > > Thank you again for giving us the opportunity to clarify this point. We also remain available to answer any questions you may have, so, please let us know.

---

> > > > ### Comment · Reviewer_PzPS · 2025-11-25
> > > >
> > > > Thanks a lot for the detailed answers. I updated my score. I just saw a prior work online that you might need to cite https://openreview.net/forum?id=qjxMqNK82L, though it does not count in my assessment.
> > > >
> > > > I'm still open to further improving my score after reading the new version of the paper, other reviewers' comments, and your answers.

---

> > > > > ### Author Response · Authors · 2025-11-26
> > > > > **Official Comment by Authors**
> > > > >
> > > > > We sincerely thank you for your thoughtful follow-up comments and for pointing us to related work, Mehrbod et al.
> > > > >
> > > > > In the revised manuscript, we have
> > > > >
> > > > > (i) added Appendix A.3 with the full random-head analysis (showing that scaling and preventative fine-tuning on off-circuit heads fails to match ASGuard’s safety–utility trade-off),
> > > > >
> > > > > (ii) clarified in Appendix A.2.2 that our absolute ASR values may differ from prior work because we evaluate all defenses under a unified pipeline with different base models and judges, and
> > > > >
> > > > > (iii) incorporated the suggested related work into our discussion of circuit-level safety, using it as complementary evidence that targeted, circuit-derived heads are indeed mechanistically meaningful.
> > > > >
> > > > > We have reflected all of these changes in the new version and would be very happy to provide any further details if helpful.

---

> ### Author Response · Authors · 2025-11-18
> **Response to Reviewer PzPS (Part 2/3)**
>
> >Representation Engineering (Turner et al., 2023) is cited but not compared. Prompt-based defenses (system prompts, few-shot examples) and Input filtering or paraphrasing are not evaluated. Authors should also compare their method against RFA (Yu et al, 2024) "refusal feature adversarial training" and Gradient Cuff.
>
> Our primary focus in this work is on **weight-level alignment methods** (post-training on model parameters), so we chose baselines that are directly comparable in this space: SFT, DPO, representation-level methods (RepE, RepBend, Circuit Breaker). Prompt-based defenses (system prompts, few-shot exemplars, paraphrasing, input filtering) operate at *inference time* and can in principle be layered on top of *any* aligned model, including ASGuard, SFT, or RepBend. For this reason, we view them as complementary rather than directly competing techniques.
>
> That said, we agree that stronger representation-level comparisons are important, and we have added additional experiments on **Llama-3.1** with **Representation Engineering** and **Gradient Cuff** in our updated results. Across these runs, ASGuard still achieves lower ASR and a better safety–utility trade-off (Overall score) than both Representation Engineering and Gradient Cuff, reinforcing that the circuit-guided training provides an additional benefit beyond these approaches.
>
> | Method | Past Tense ASR (↓) | OR-Bench Toxic (↑) | OR-Bench Hard (↓) | MMLU (↑) |
> | --- | --- | --- | --- | --- |
> | **Llama-3.1-8B-Instruct** | 42 | 88.5 | 28.9 | 68.2 |
> | Head Ablation | 29 | 86.7 | 34.7 | 65.2 |
> | SFT (5/95) | 21 | 94.1 | 50.8 | 67.8 |
> | SFT (30/70) | 3 | 91.9 | 80.3 | 67.7 |
> | DPO | 38 | 90.2 | 33.2 | 68.0 |
> | RepE | 41 | 87.9 | 29.7 | 68.3 |
> | CB | **0** | 85.1 | 84.8 | 68.2 |
> | Gradient Cuff | 51 | 40.9 | 9.02 | 32.8 |
> | RepBend | 11 | 96.1 | 77.9 | 68.2 |
> | Only Scaling (Ours) | 13 | 96.9 | 66.2 | 64.3 |
> | ASGuard (Ours) | 8 | 96.4 | 66.8 | 68.2 |
>
> **hyperparams of RepE: build refusal vector with last layer, scaling factor alpha 3.0*
>
> **hyperparams of Gradient Cuff: using 50 benign prompts, search model specific gc threshold (for Llama3.1, 31.776). other settings follow basic params suggested by original paper: p_times 10, sample times 10, mu 0.02*
>
> Regarding RFA and ReFAT (Yu et al., 2024) [3]; to the best of our knowledge there is no official open-source implementation, and a faithful reimplementation would require substantial engineering effort. We therefore leave a direct empirical comparison to ReFAT as future work and will state this explicitly in the revised version.
>
> >Furthermore, there seem to be some issues with the RepBend comparison: RepBend is described as "recent SoTA" but it's surprising that it doesn't consistently outperform simpler methods. Some results show RepBend > ASGuard on specific metrics (e.g., Llama Toxic: 96.1 vs 96.4).
>
> We appreciate the chance to clarify this point. In our Llama-3.1 results, **RepBend** achieves 96.1 on OR-Bench-Toxic, while **ASGuard reaches 96.4**; since higher OR-Bench-Toxic is better, ASGuard is actually *slightly* stronger on this metric. Our description of RepBend as “recent SoTA” refers to its status in the *prior literature*, not to its performance relative to ASGuard in our experiments.
>
> More broadly, our results confirm that RepBend is indeed a strong and robust baseline: it substantially reduces ASR and attains a good Overall score, validating it as a competitive method. ASGuard, however, consistently pushes the safety–utility frontier further (lower ASR and higher Overall) on Llama-3.1 and other models, indicating that our circuit-guided preventative fine-tuning improves on an already strong SoTA representation-engineering baseline.
>
> >What are the computational costs relative to standard fine-tuning? Can you ablate on detection dataset size?
>
> ASGuard introduces an additional *one-time* cost for circuit construction and scaling-vector training on a **small detection dataset** (100 JBB prompts with 20 reformulations each). Once the vulnerable circuit is identified and the scaling vectors are trained, **preventative fine-tuning itself is a standard supervised fine-tuning step on the refusal dataset**, comparable in cost to light SFT or DPO (we run 1 epoch with FSDP on two H100 GPUs). Crucially, the resulting scaling can be fused into the attention output projection with **no extra inference cost**.
>
> We have not yet performed a systematic ablation over the size of the detection dataset; however, our current results show strong safety–utility gains with only 100 base prompts, suggesting that ASGuard is relatively sample-efficient. We will add a brief discussion of computational cost and explicitly mention dataset-size ablation as future work.
>
> ---
>
> [3] Robust LLM safeguarding via refusal feature adversarial training, Yu et al., 2024

---

> ### Author Response · Authors · 2025-11-18
> **Response to Reviewer PzPS (Part 3/3)**
>
> >If you train defenses for multiple attack types, do they interfere? Can one set of scaling vectors handle multiple attacks?
>
> In our current experiments, we train **separate** scaling vectors for each target attack (tense, GCG, LogiBreak) and do not yet combine them into a single multi-attack scaling vector. However, we observe that a model trained for tense jailbreak robustness already exhibits strong OOD robustness to unseen GCG and LogiBreak attacks, suggesting that preventative fine-tuning encourages reuse of safer refusal circuits rather than isolated, attack-specific heuristics. When we explicitly use the *tense-robust* ASGuard Llama and evaluate it on unseen GCG (base model ASR 15%) and LogiBreak (base model ASR 30%) attacks, it achieves ASR 1% and 15% respectively without any additional training. You can also see the result in revised §5.3.
>
> Studying how multiple scaling vectors interact—e.g., by composing them or learning a joint multi-attack scaling vector—is an exciting direction, but outside the scope of this work. We will clarify this limitation and point to multi-attack defenses as future work.
>
> >Why does preventative fine-tuning work? Can you provide formal analysis or at least deeper mechanistic explanation?
>
> Following similar findings of Chen et al [4], §3.3 and §6 offer a mechanistic explanation that we will emphasize more clearly. Preventative fine-tuning treats the learned scaling vectors as fixed “steering directions” integrated into the model: during fine-tuning, the vulnerable pathway is consistently down-weighted, increasing the optimization cost of routing refusal behavior through the original jailbreak circuit. This forces the model to discover alternative, safer computation paths.
>
> Our linear probe analysis supports this picture: after preventative fine-tuning, some of the formerly heads become *better* tense detectors (their probe accuracy increases), indicating that their representations have been sharpened while their role in the harmful pathway has been re-routed. We will consolidate this explanation in §6 and more clearly connect it to prior work on steering vectors and representation-level alignment.
>
> ---
>
> Thank you for your feedback, which has greatly enhanced our work. We believe these revisions address your comments effectively.
>
> ---
>
> [4] Persona vectors: Monitoring and controlling character traits in language models, Chen et al., 2025

---

> > ### Author Response · Authors · 2025-11-24
> > **Follow-up on Author Response**
> >
> > Dear Reviewer PzPS,
> >
> > Thank you for your thoughtful and constructive feedback on our manuscript. We have thoroughly addressed your comments and submitted a revised version for your consideration. We would greatly appreciate it if you could review our responses and the updated manuscript at your earliest convenience. We understand the demanding nature of the review process and are grateful for the time and effort you are dedicating to our work.

---

### Official Review · Reviewer_vndG · 2025-11-04

**Soundness:** 3
**Presentation:** 3
**Contribution:** 3
**Rating:** 6
**Confidence:** 2

**Summary:**

The paper proposes ASGUARD, an activation-scaling method to mitigate targeted jailbreaks like tense-based attacks. It identifies vulnerable attention heads via circuit attribution, scales their activations to suppress unsafe behavior, and uses a short preventative fine-tuning phase to internalize the fix. Experiments across models show lower attack success rates and limited over-refusal while maintaining utility. The approach has demonstrated a practical and interpretable defense to the specific type of attack.

**Strengths:**

1. The paper targets a concrete jailbreak based on tense-based rephrasing and provides an interpretable fix.
2. The activation-scaling intervention is lightweight, requiring only head-level scaling vectors rather than any weight updates.
3. The preventative fine-tuning shows that it is feasible to transfer the scaling effect into model parameters, achieving Pareto-optimal results under this compounded method.
4. The experiments generalize across architectures and model sizes, showing that the internal behavior is replicable and the robustness holds consistently across models.

**Weaknesses:**

1. The attribution step that identifies “vulnerable heads” is largely heuristic. It would be beneficial to demonstrate that the selected heads are causally necessary for the jailbreak behavior. Additional analyses, such as head ablation or random-head controls, would make the causal interpretation more convincing.

2. The method is designed specifically for tense-based jailbreaks, and it is unclear whether the same workflow would generalize to other jailbreak categories. While the paper does not overclaim, the scope still appears limited.

**Questions:**

1. Have you verified whether removing or scaling different random sets of heads produces similar reductions in attack success?

3. Could ASGUARD be extended to other types of jailbreaks, and if so, how would “vulnerable heads” be defined in those cases?

3. How well does the preventative fine-tuning generalize to OOD data?

---

> ### Author Response · Authors · 2025-11-18
> **Response to Reviewer vndG (Part 1/2)**
>
> We greatly appreciate your valuable feedback, which has been instrumental in improving our paper.
>
> ---
>
> >It would be beneficial to demonstrate that the selected heads are causally necessary for the jailbreak behavior. Additional analyses, such as head ablation or random-head controls, would make the causal interpretation more convincing.
>
> >Have you verified whether removing or scaling different random sets of heads produces similar reductions in attack success?
>
> We fully agree that demonstrating causal necessity is crucial, and our analysis with linear probe makes this evidence more explicit.
>
> We already provide both causal and mechanistic evidence that the selected heads are responsible for tense jailbreaks. In §3.1, ablating only the EAP-IG–identified vulnerable heads reduces past-tense ASR by 4–13%, whereas ablating the same number of random heads changes ASR by only ≈1–2%, indicating that our heads have a disproportionate causal impact on the attack.
>
> In §6, we further reconstruct the circuits after preventative fine-tuning and perform linear-probe analysis: most of the original vulnerable heads no longer appear in the reconstructed jailbreak circuits, yet their tense-probe accuracy *increases* (e.g., L10H19 from 71.88% to 73.44%), showing that ASGuard does not erase their function but sharpens tense representations while realigning how they are used in the circuit.
>
> Additional internal experiments with Temporal Heads [1]—conceptually related but not identified by our circuit analysis—show that ablating or scaling them barely affects ASR or utility, supporting that our heads are not just “any tense-like heads” but specifically those that causally drive the vulnerability.
>
> We will clarify and more explicitly describe the random-head ablation as a control in the further revision.
>
> >it is unclear whether the same workflow would generalize to other jailbreak categories.
>
> >Could ASGUARD be extended to other types of jailbreaks, and if so, how would “vulnerable heads” be defined in those cases?
>
> To address this concern, we explicitly applied the **same three-stage workflow**—circuit discovery, activation scaling, and preventative fine-tuning—to two additional jailbreak families: **GCG** (adversarial suffix-based attacks) [2] and **LogiBreak** (logical-form jailbreaks) [3].
>
> 1. **Extending circuit discovery beyond tense jailbreaks.**
>
> Using the same EAP-IG circuit-construction procedure as for tense jailbreaks, we build circuits for successful vs. failed GCG and LogiBreak attacks on Llama-3.1. We again contrast “success” and “failure” circuits (analogous to “False-to-True” vs. “Always-False”) and identify a small set of GCG- and LogiBreak-vulnerable heads.
>
> | Model | List of GCG Vulnerable Heads | List of LogiBreak Vulnerable Heads |
> | --- | --- | --- |
> | Llama-3.1-8B-Instruct | L0H3, L1H24, L30H1, L31H5 | L24H27, L28H13 |
>
> 2. **Applying activation scaling + preventative fine-tuning to GCG and LogiBreak.**
>
> We then train channel-wise scaling vectors and perform preventative fine-tuning using the same Identify-then-Scale protocol as in the tense setting. On **GCG**, ASGuard reduces the ASR from **15% to 1%** and achieves the highest R-Score (76.0) and Overall score among selected baselines, outperforming both SFT (30/70) and RepBend.
>
> On **LogiBreak**, ASGuard attains a **moderate ASR (13%)** while SFT and RepBend collapse their R-Scores to 0 (due to severe drops on OR-Bench Toxic), again yielding the best Overall safety–utility trade-off.
>
> *(a) Results on GCG (Zou et al., 2023) Benchmark*
>
> | Method | GCG ASR (↓) | OR-Bench Toxic (↑) | OR-Bench Hard (↓) | MMLU (↑) | R-Score (↑) | Overall (↑) |
> | --- | --- | --- | --- | --- | --- | --- |
> | Llama-3.1-8B-Instruct | 15 | 88.5 | 28.9 | 68.2 | – | – |
> | SFT (30/70) | 2 | 66.1 | 2.27 | 67.3 | 13.8 | 13.4 |
> | RepBend | 5 | 94.3 | 50.4 | 68.1 | 73.3 | 41.7 |
> | ASGuard (Ours) | **1** | 96.7 | 59.5 | 68.3 | **76.0** | **45.0** |
>
> *(b) Results on LogiBreak (Peng et al., 2025) Benchmark*
>
> | Method | LogiBreak ASR (↓) | OR-Bench Toxic (↑) | OR-Bench Hard (↓) | MMLU (↑) | R-Score (↑) | Overall (↑) |
> | --- | --- | --- | --- | --- | --- | --- |
> | Llama-3.1-8B-Instruct | 30 | 88.5 | 28.9 | 68.2 | – | – |
> | SFT (30/70) | **0** | 59.3 | 1.59 | 66.7 | 0* | 15.0 |
> | RepBend | 13 | 68.7 | 78.9 | 68.2 | 0* | 8.5 |
> | ASGuard (Ours) | 13 | 97.1 | 64.9 | 68.1 | **74.7** | **45.8** |
>
> ---
>
> [1] Does Time Have Its Place? Temporal Heads: Where Language Models Recall Time-specific Information
>
> [2] Universal and transferable adversarial attacks on aligned language models, Zou et al., 2023
>
> [3] Logic Jailbreak: Efficiently Unlocking LLM Safety Restrictions Through Formal Logical Expression, Peng et al., 2025

---

> > ### Comment · Reviewer_PzPS · 2025-11-25
> > **What about random drop + scaling + preventive scaling?**
> >
> > Did you evaluate the impact/ablation of the choice of heads (random drop) in the whole pipeline?

---

> > > ### Author Response · Authors · 2025-11-25
> > > **Follow-up on question about random drop + scaling + preventative fine-tuning**
> > >
> > > To directly address your follow-up question, we ran a full-pipeline control experiment where we replaced our EAP-IG–identified vulnerable heads with a set of heads that never appear in *any* of the tense circuits. On Llama-3.1-8B-Instruct, we randomly selected 10 attention heads that do not occur in either the False-to-True or Always-False circuits from §3: L16H6, L26H4, L18H19, L24H4, L23H24, L23H15, L15H30, L30H18, L4H2, L14H5.
> > >
> > > We then applied **the same two interventions** as in ASGuard:
> > >
> > > (i) channel-wise activation scaling (“Random Scaling”) and
> > >
> > > (ii) scaling followed by preventative fine-tuning (“Random PFT”),
> > >
> > > keeping all hyperparameters identical to our main tense-jailbreak setup. The results are:
> > >
> > > | Method | Past Tense ASR (↓) | OR-Bench Toxic (↑) | OR-Bench Hard (↓) | MMLU (↑) |
> > > | --- | --- | --- | --- | --- |
> > > | **Llama-3.1-8B-Instruct** | 42 | 88.5 | 28.9 | 68.2 |
> > > | SFT (5/95) | 21 | 94.1 | 50.8 | 67.8 |
> > > | SFT (30/70) | 3 | 91.9 | 80.3 | 67.7 |
> > > | Random Scaling | 25 | 91.3 | 43.3 | 67.4 |
> > > | Random PFT | 5 | 98.9 | 89.0 | 68.2 |
> > > | Only Scaling (Ours) | 13 | 96.9 | 66.2 | 64.3 |
> > > | ASGUARD (Ours) | 8 | 96.4 | 66.8 | 68.2 |
> > >
> > > Here, several observations follow.
> > >
> > > 1. **Random Scaling is not competitive with circuit-based scaling.**
> > >
> > >     Random Scaling does reduce past-tense ASR (42 → 25), but it is weaker than both our circuit-based Only Scaling (13 ASR) and ASGuard (8 ASR), while providing only modest gains on OR-Bench Toxic (88.5 → 91.3) and essentially no benefit over SFT(5/95). This suggests that simply down-weighting arbitrary, off-circuit heads does not yield the strong, targeted robustness we obtain when scaling heads that lie on the tense-jailbreak pathway.
> > >
> > > 2. **Random PFT achieves low ASR only via extreme over-refusal.**
> > >
> > >     Random PFT pushes ASR down to 5, but at the cost of **severe** over-refusal: OR-Bench Hard jumps from 28.9 to 89.0 (even higher than SFT(30/70), which reaches 80.3), indicating that the model is close to “always refuse.” In contrast, ASGuard attains a comparable ASR of 8 while maintaining substantially lower OR-Bench Hard (66.8 vs 89.0). Thus, the random-head pipeline reduces ASR primarily by globally suppressing helpful behavior, not by selectively mitigating the jailbreak mechanism.
> > >
> > > 3. **Head choice matters for the whole pipeline, not just ablation.**
> > >
> > >     These controls show that applying the same recipe to arbitrary heads does **not** reproduce the safety–utility trade-off achieved by ASGuard. The full pipeline is most effective when the scaled heads are those that circuit analysis identifies as participating in the tense jailbreak. This supports our claim that the attribution step is doing substantive causal work, rather than merely acting as a generic regularizer.
> > >
> > > Thank you for giving us the opportunity to clarify this point. We will incorporate these additional analyses into the revised manuscript. Please let us know if you have any further questions or concerns.

---

> > > > ### Comment · Reviewer_PzPS · 2025-11-25
> > > > **Thanks!**
> > > >
> > > > I thank the authors for their quick reply and the new experiments. It answered my concerns.

---

> ### Author Response · Authors · 2025-11-18
> **Response to Reviewer vndG (Part 2/2)**
>
> 3. **Cross-model generalization.**
>
> Beyond existing models, we further apply ASGuard to **OLMo-2-1124-7B-Instruct**, observing consistently strong ASR reductions and high R-Scores in Table 2. For instance,on Olmo2,  ASGuard pushes the safety–utility frontier compared to all baselines.
>
> The Pareto-front plots in Figure 2 show that, across all four base models, ASGuard consistently lies on or near the safety–utility frontier, combining strong ASR reduction with high R-Scores. This frontier behavior holds not only for tense jailbreaking but also when we consider the additional GCG and LogiBreak results, indicating that the workflow generalizes beyond the specific linguistic perturbation considered initially.
>
> The result of Olmo2 with relative robustness is like below:
>
> | Method | ASR (↓) | OR-Bench Toxic (↑) | OR-Bench Hard (↓) | MMLU (↑) | R-Score (↑) | Overall (↑) |
> | :--- | ---: | ---: | ---: | ---: | ---: | ---: |
> | **OLMo-2-1124-7B-Instruct** | 28 | 92.5 | 43.5 | 60.5 | – | – |
> | Head Ablation | 22 | 92.5 | 45.1 | 60.1 | 65.5 | 43.5 |
> | FT (5/95) | 21 | 94.8 | 57.9 | 59.0 | 67.6 | 46.3 |
> | FT (30/70) | **8** | 99.6 | 91.8 | 58.4 | 68.6 | 70.0 |
> | DPO | 25 | 93.2 | 48.3 | 60.5 | 66.9 | 38.8 |
> | CB | 20 | 92.3 | 43.1 | 60.5 | 66.0 | 47.3 |
> | RepBend | 23 | 92.0 | 42.8 | 60.5 | 64.9 | 41.4 |
> | Only Scaling (Ours) | 17 | 92.8 | 48.8 | 59.5 | 64.3 | 51.8 |
> | **ASGuard (Ours)** | 9 | 97.5 | 69.2 | 60.6 | **73.7** | **70.8** |
>
> >How well does the preventative fine-tuning generalize to OOD data?
>
> We evaluate OOD robustness along both broad-distribution and adversarial axes.
>
> First, OR-Bench-Toxic is distributionally distinct from tense jailbreak; yet Table 2 shows that ASGuard substantially improves OR-Bench-Toxic while keeping OR-Bench-Hard and MMLU stable, leading to the highest R-Score and Overall score across all methods on Llama-3.1.
>
> Second, in revised §5.3 we explicitly use the *tense-robust* ASGuard model (trained only on tense jailbreak data) and evaluate it on unseen GCG (base model ASR 15%) and LogiBreak (base model ASR 30%) attacks, achieving ASR 1% and 15% respectively without any additional training, which directly demonstrates OOD generalization to new jailbreak families. Conceptually, this arises because preventative fine-tuning is performed while the vulnerable pathway is down-weighted by the scaling vector, forcing the model to learn alternative, safer refusal circuits that are not only specific to a single template.
>
> ---
>
> We hope these responses address your concerns. Your feedback has been essential in improving our work, and we look forward to refining it further. Thank you for your review.

---

> > ### Author Response · Authors · 2025-11-24
> > **Follow-up on Author Response**
> >
> > Dear Reviewer vndG,
> >
> > We are writing to kindly ask if you have had a chance to review our response to your comments. We have made significant updates to the manuscript based on your valuable suggestions, which we believe have strengthened the paper. As the discussion period is progressing, we would be happy to answer any further questions or provide additional clarifications if needed. Thank you for your time and contribution to improving our work.

---

### Author Response · Authors · 2025-11-27
**General Response to All Reviewers**

Dear Reviewers and Chairs,

We sincerely thank the reviewers for their insightful and constructive feedback. We are encouraged that the reviewers commended the **strong mechanistic foundation** of our work (@PzPS, @KbDQ) and recognized our "Preventative Fine-Tuning" as a **novel defense paradigm** that effectively transfers temporary scaling effects into permanent model parameters (@vndG, @PzPS, @HMBL).

Specifically, reviewers highlighted that our approach achieves **Pareto-optimal safety–utility trade-offs** (@vndG, @PzPS) and is **well-generalized across different model architectures** (@vndG). We also appreciate the feedback that the paper is **clearly structured and easy to follow** (@HMBL).

Building on these strengths, we have extensively revised the manuscript to incorporate new experiments and analyses that address the reviewers' specific suggestions. Below, we summarize the major improvements.

---

## 1. Generalization to New Jailbreak Families (GCG & LogiBreak)

A common concern raised by reviewers was whether ASGuard is specific only to “tense jailbreaks.” To demonstrate the generality of our workflow, we applied the exact same three-stage pipeline—(1) EAP-IG circuit discovery, (2) activation scaling, and (3) preventative fine-tuning—to two additional distinct attack categories: GCG (adversarial suffix) and LogiBreak (logical manipulation) (@vndG, @PzPS).

As shown in the new results (Section 5 and Appendix), ASGuard successfully identifies attack-specific vulnerable heads and significantly reduces Attack Success Rate (ASR) while maintaining high utility, outperforming strong baselines like RepBend and SFT (@PzPS, @KbDQ).
We also verified via TenseLoC that ASGuard **preserves fine-grained tense understanding** (@HMBL).

Furthermore, we found that the model trained only on tense jailbreaks exhibits strong Out-Of-Distribution (OOD) robustness against unseen GCG and LogiBreak attacks, suggesting that ASGuard encourages the model to learn robust refusal circuits rather than memorizing specific templates (@vndG, @PzPS).

## 2. Causal Necessity via Random-Head Controls

To verify that our selected heads are mechanistically causal rather than arbitrary, we conducted a rigorous control experiment using Random Scaling and Random Preventative Fine-Tuning (PFT) on heads disjoint from the jailbreak circuit (@vndG).

- **Random Scaling**: Failed to reduce ASR effectively compared to our targeted scaling (@vndG).

- **Random PFT**: Reduced ASR but caused severe over-refusal (OR-Bench Hard score dropped drastically), behaving like a generic suppression (@HMBL).

- **ASGuard**: Achieved the best safety–utility trade-off. This confirms that targeting the specific heads identified by our circuit analysis is essential for "surgical" repair without compromising general capabilities (@vndG, @KbDQ).

## 3. Expanded Baselines and Evaluation

We have significantly expanded our comparative evaluation to include:

- **Additional Representation Engineering Methods**: We added comparisons with Representation Engineering (RepE) and Gradient Cuff. ASGuard consistently achieves a superior Pareto frontier (@PzPS).

- **Parameter-Efficient Fine-Tuning**: We tested SFT with LoRA. While LoRA mitigated catastrophic forgetting, it failed to reduce ASR effectively compared to ASGuard, highlighting that our method's efficacy comes from targeted intervention, not just parameter efficiency (@HMBL).

- **Unified Evaluation Pipeline**: We clarified discrepancies regarding baselines like RepBend by evaluating all methods under a unified, rigorous pipeline (same prompts, judges, and base model Llama-3.1), ensuring a fair comparison (@PzPS, @HMBL).

## 4. Computational Efficiency & Mechanistic Insight

- **Efficiency**: We clarified that the cost of circuit discovery is a one-time investment (approx. 30 mins) and that the subsequent scaling and fine-tuning steps are extremely lightweight (seconds to minutes) compared to full SFT or DPO runs (@HMBL, @PzPS).

- **Mechanism**: We reinforced the mechanistic explanation in Section 6. Preventative fine-tuning works by "training under handicap"—down-weighting the vulnerable pathway during training forces the optimizer to re-route the refusal computation through safer, alternative circuits (@KbDQ, @vndG).

---

We believe these revisions comprehensively address the reviewers’ concerns and demonstrate that ASGuard is a robust, generalizable, and mechanistically grounded defense framework.

We also deeply appreciate the Committee’s dedication amidst recent disruptions. While we regret that the suspension curtailed our positive ongoing discussions before final confirmation, we respectfully ask the upcoming AC to consider our comprehensive rebuttal which we believe fully resolves the raised concerns.

---

### Meta-Review · Area_Chair_9qK3 · 2026-01-18

**Summary:**

This paper proposes ASGuard, a mechanistically informed defense against targeted jailbreaks. Overall, reviewers agree the approach is technically competent and the mechanistic framing is a key strength. A major concern among all reviewers is that the method is only designed for tense jailbreaking. In the rebuttal and revision, the authors substantially strengthened the paper with new generalization experiments (GCG and LogiBreak), random head and ablation controls, and expanded baseline comparisons under a more unified evaluation pipeline. I recommend acceptance.

**Reviewer Concerns:**

* All reviewers raised concerns that the method appeared tailored to tense jailbreaking. In response, the authors applied the same three-stage workflow to two additional jailbreak methods, GCG and LogiBreak, reporting large ASR reductions while maintaining utility metrics.

* Reviewer vndG asked if the selected heads are causally necessary for the jailbreak behavior. The authors added targeted evidence by conducting additional ablation studies on the choice of heads to scale. These experiments make the causal interpretation more convincing.

* Reviewers PzPS and KbDQ requested stronger and broader baseline comparisons. The authors added comparisons with RepE and RepE. ASGuard consistently achieves a superior Pareto frontier.

* Reviewer PzPS questioned whether the ASR drop could be explained by increased over-refusal. The authors addressed this by analyzing safety-utility trade-offs with OR-Bench and showing ASGuard achieves comparable ASR reduction with substantially less over-refusal than generic fine-tuning baselines.

**Reviewer Scores:**

Based on the discussion and the additional experiments provided in the revision, the main concerns raised by the reviewers have been substantially addressed. As a result, reviewer scores are likely to improve:
* Reviewer vndG is likely to increase the score to 8
* Reviewer PzPS is likely to increase the score to 6
* Reviewer KbDQ is likely to increase the score to 6
* Reviewer HMBL is likely to keep the positive score 6.

---

### Decision · Program_Chairs · 2026-01-26

Accept (Poster)